# Fully Convolutional Network for the Semantic Segmentation of Medical Images: A Survey

**DOI:** 10.3390/diagnostics12112765

**Published:** 2022-11-11

**Authors:** Sheng-Yao Huang, Wen-Lin Hsu, Ren-Jun Hsu, Dai-Wei Liu

**Affiliations:** 1Institute of Medical Science, Tzu Chi University, Hualien 97071, Taiwan; 2Department of Radiation Oncology, Hualien Tzu Chi General Hospital, Buddhist Tzu Chi Medical Foundation, Hualien 97071, Taiwan; 3Cancer Center, Hualien Tzu Chi Hospital, Buddhist Tzu Chi Medical Foundation, Hualien 97071, Taiwan; 4School of Medicine, Tzu Chi University, Hualien 97071, Taiwan

**Keywords:** semantic segmentation, medical image processing, deep learning, fully-convolutional network

## Abstract

There have been major developments in deep learning in computer vision since the 2010s. Deep learning has contributed to a wealth of data in medical image processing, and semantic segmentation is a salient technique in this field. This study retrospectively reviews recent studies on the application of deep learning for segmentation tasks in medical imaging and proposes potential directions for future development, including model development, data augmentation processing, and dataset creation. The strengths and deficiencies of studies on models and data augmentation, as well as their application to medical image segmentation, were analyzed. Fully convolutional network developments have led to the creation of the U-Net and its derivatives. Another noteworthy image segmentation model is DeepLab. Regarding data augmentation, due to the low data volume of medical images, most studies focus on means to increase the wealth of medical image data. Generative adversarial networks (GAN) increase data volume via deep learning. Despite the increasing types of medical image datasets, there is still a deficiency of datasets on specific problems, which should be improved moving forward. Considering the wealth of ongoing research on the application of deep learning processing to medical image segmentation, the data volume and practical clinical application problems must be addressed to ensure that the results are properly applied.

## 1. Introduction

Medical imaging has long functioned as an assistive means for diagnosis and treatment. Advancements in technology have increased the types and qualities of medical images. Lesion detection is one of the primary objectives of medical imaging, as the size and location of lesions are often directly associated with a patient’s diagnosis, treatment, and prognosis. Previously, the size and location of lesions were determined by radiologists through medical image examination. At best, the instruments and software used were only able to enhance the image quality by adjusting the brightness and contrast features to facilitate better observation. Since the development of computer vision algorithms, however, researchers have begun to utilize these algorithms in the field of medical imaging [1].

As the core of deep learning, convolutional neural networks (CNN) (Figure 1) have a considerably long development history [2]. However, due to hardware-related limitations, it was only in the 2010s that breakthroughs in the effectiveness of CNNs were made [3]. Meanwhile, deep learning models that meet specific targets have gradually been proposed, from classification models, to object detection, to object segmentation. Consequently, advancements such as the detection of lung diseases through X-ray [4], the detection of lesion locations [5], and segmentation have been applied in medical imaging. Due to improvements in model performance, deep learning models have exhibited diagnostic capabilities approximating those of clinical physicians, based on the inference of specific datasets. However, in traditional CNN models, there are limitations to segmentation. One of them is about the features extracted from the models. When using a smaller kernel, the features will become more local to the original image. Global information, such as location, may be lost. However, when using a larger kernel, the context of features may decrease [6]. Another limitation is the data availability of biomedical segmentation tasks. Due to privacy in the medical profession, the medical data volume is often small when compared with the data volume in other fields [6]. New models have been developed to solve these problems. Another solution is data augmentation.

In this article, we will discuss the models proposed for semantic segmentation and data augmentation techniques, as well as their current applications on clinical data.

## 2. Fully Convolutional Network (FCN)

One of the most-selected models for segmentation tasks is FCN (Figure 2). The difference between FCNs and traditional convolutional neural networks is that the last layer is not a fully connected layer that enables the model to integrate information. Instead, the final number of output channels is modified through convolutional networks. The main benefit of this approach is that the model has no restrictions from the full connection layer, so the size of the input can be flexible. Table 1 shows the models used for semantic segmentation tasks.

The U-Net is an example of an FCN (Figure 3) [6]. The U-Net leverages the extraction features of convolutional networks, in which the upsampling layer and concatenation are used to compare the features of the top layer while simultaneously retaining the features of the bottom layer. The model is thereby able to detect the detailed and general features of objects, thus detecting the location of objects. The U-Net can be referred to as a pixel-level image classification method, in which the segmentation is performed through pixel-wise classification. The U-Net architecture is briefly described as follows:

The descending part is also known as the encoding region. It is composed of units called convolutional blocks, which include convolutional and pooling layers and are akin to those of traditional convolutional neural networks and sometimes may include batch normalization layers. Before entering a pooling layer, features are filtered through an activation function that determines whether the extracted features should be transferred to the next layer. The most common activation functions are ReLu and sigmoid, expressed as the following equations:

(1)$$S\left(x\right)=\frac{1}{1+e^{-x}}\;<\mathrm{sigmoid}>$$
where *x* is the feature extracted from the CNN kernel after the weight is summed, and e is exponential.
(2)$$f\left(x\right)=\left\{\begin{array}{c} 0,\;x<0\\ w^{T}x+b,\;x\geq 0 \end{array}\right.\;<\mathrm{ReLu}>$$
where *x* is the feature extracted from the CNN kernel, *w* is the weight, and *b* is the bias.Before passing to the next convolutional blocks, the features will be retained. While there is no limit on the number of layers in the descending parts, it is usually set at four.

2.The ascending part is also known as the decoding region. One of the keys of its architecture is that the size of a feature map is upscaled after passing through the upsampling layer. Another key is that, after upsampling, the feature will be concatenated with the feature retained at the corresponding level in the descending part. The ascending part mainly consists of alternating upsampling and pooling layers. In the upsampling layer, the following methods are usually used to upscale the features:
Nearest neighbor interpolation: The value assigned is equal to that of the nearest pixel.Bilinear interpolation: The value is obtained through bilateral linear interpolation.Cubic interpolation: Third-degree polynomials are used to obtain values. Batch normalization, which can still be used, undergoes activation function processing before entering the upsampling layer.3.Output layer: Before the result export, the feature reaching the top of the ascending part is processed. The layer is typically a single convolutional layer and usually makes predictions pixel by pixel. The Softmax function can be applied to the layer, as it generates the probability distribution of the classification. If the length and width of an original input image are x and y, respectively, then the output tensor size will be (x,y,c), with c representing the number of classes.4.The loss function: Cross-entropy or the dice coefficient is often used. The respective math equations are as follows:

(3)$$H\left(p,q\right)=-\sum_{x}p\left(x\right)\log q\left(x\right)\;<\mathrm{cross}\;\mathrm{entropy}>$$
where *p*(*x*) is the ground truth probability distribution, and *q*(*x*) is the probability distribution of the result predicted by the model.
(4)$$s=\frac{2\left|X\cap Y\right|}{\left|X\right|+\left|Y\right|}\;<\mathrm{dice}\;\mathrm{coefficient}>$$
where *X* represents the result predicted by the model, and *Y* is the ground truth.

### 2.1. Modified U-Net Models

#### 2.1.1. Modifying Encoder

The descending part of the U-Net architecture is structurally like the feature extraction process of CNN; hence, the feature extraction part of some CNN classification frameworks, such as VGG or ResNet, is used to replace the convolutional block of descending part. Jakhar et al. [13] applied a simple U-Net structure for segmentation and obtained considerably decent results. Abedalla et al. [8] applied the ResNet model to complete segmentation tasks.

#### 2.1.2. Application of Residual Block

The residual block originates from ResNet (Figure 4a) [14,15]. The core concept is identity mapping. In the past, due to deep learning neural network stacking with convolutional layers, there is gradient loss during backpropagation, and the weights cannot be renewed effectively at the top layer. An identity map has two paths: one extracts features from the CNN layer-by-layer, as traditional CNN layers do, while the other skips those CNN layers and sums the feature extracted from the previous path before entering the next step. The latter path is also known as a skip connection because it does not pass through the stacked CNN layers. Features that pass through a skip connection are retained and extracted in the lower layers. During backpropagation, the skip connection can keep the gradient from vanishing or being lost.

The encoder–decoder architecture of U-Net is also full of stacked CNN layers in those convolutional blocks. Even though the skip connection already existed in the original U-Net and allowed features to be fully or partially retained before concatenation, some studies have attempted to apply skip connections to other parts of the framework. For example, Isensee et al. [16] modified the original U-Net with skip connections. In the proposed model, there are two paths after reaching the next level layer in the descending part. One path goes through the upsampling, concatenation, and CNN process as the original U-Net did, while the other directly sums up the features of the upper CNN output. In this way, the sensitivity of the U-Net increases.

There was a study that exclusively examined the importance of partial skip connections [17]. In this context, a long skip connection crosses between the encoder–decoder architecture like a concatenated skip connection in the U-Net, while a short skip connection is similar to a residual block in a CNN block. The study found that short skip connections are more stable and efficient for weight updates, as they prevent the weight from hovering around extreme values so that the gradient can descend in a stable manner.

#### 2.1.3. Application of Squeeze-and-Excitation Module

The squeeze-and-excitation (SE) (Figure 4b) module was first proposed by Hu et al. [18] in 2017. The idea of this module is that each channel may add a different contribution in a feature map. Therefore, weights are first submitted by each channel as a representation of their contribution, then they are multiplied by the feature map. In this way, the signals of more contributive channels are enhanced while others are suppressed.

Wang et al. [9] applied SE modules to both spatial and channel feature maps in U-Net by using DenseNet as a backbone. The mean pixel-wise accuracy and Dice similarity coefficient (DSC) were both over 90%.

#### 2.1.4. Application of Dilated Convolution Network

A dilated convolution network (also known as an atrous convolution) (Figure 4c) expands the convolution receptive field while retaining the number of convolution kernels. The mathematic expression is as follows:(5)$$y_{i}=\sum_{j}x_{i+rj}w_{j}$$
where *y* is the output of the dilated convolution, *x* is the input, and *w* is the kernel weight, which can be expressed as follows:(6)$$K_{d}\left(x,y\right)=\left\{\begin{array}{c} K_{o}\left(i,j\right)\;if\;x=i\cdot \alpha ,\;y=i\cdot \beta \;\\ 0\;else\; \end{array}\right.$$

A schematic diagram is shown below. The significance of this process is to increase the receptive field of the CNN kernel to maintain the features of the larger region level as the model goes deeper. Combined with U-Net, this process enables the model to have a more precise prediction at the edge of the object and increases the robustness. A larger scale of dilated CNN kernels is used for learning the location of the edge in an object while smaller dilated CNN models are used for identifying the direction of the edge. When applied to an object with blurred edges, high-rated dilated CNN kernels can determine whether an object is blurry [19] while small-rated dilated CNN kernels can identify how blurred the object is. By doing so, the model can differentiate those similar blurry objects around the target.

Xiao et al. proposed a model that combined U-Net with a dilated CNN called MSDU-Net. In the model, features will be extracted from the input by the dilated CNN before entering U-shaped architecture. As the layer goes deeper, the rate of the dilated CNN (represented by *r* in Equation (5)) increases. As a result, the receptive field increases while the size of the feature decreases so that it can be the same size as the feature extracted from the upper layer of the encoder. The feature extracted from the upper layer of the encoder and extracted from the corresponding level of the dilated CNN will be concatenated and then transferred to the next level of the encoder [19].

In addition to detecting blurry edges, a dilated CNN kernel is capable of sharpening the edges of an object. Zhou et al. [10] utilized a dilated CNN to attenuate the noise from artifacts in medical imaging. Another study [20] applied dilated CNNs solely to increase the intersection over union (IoU).

#### 2.1.5. Application of the Attention Module

The attention module (Figure 4d) was first used in deep learning models for natural language processing [21]. It figures out the correlation between the query and key to obtain the weight of the value corresponding to the key, and then it sums the weight of the value to get the attention. As a result, the attention module can capture the relationship between texts in a region or paragraph more precisely. The mathematical equation is expressed as follows:(7)$$ttention\left(Q,K,V\right)=Softmax\left(\frac{QK^{T}}{\sqrt{d_{k}}}\right)V$$

Similar concepts are now used in the computer vision field. Applying attention modules makes the model combine the correlations of regional features, generating better object edge detection results to improve performance.

Yousefi et al. [11] applied the attention module, combined with a dilated CNN, to the U-Net. Attention modules were used within the convolutional block and between the skip connections. The model was tested on a chest CT dataset for esophagus segmentation tasks, and the AUC was 0.76. Cai et al. [12] presented a similar method in which an attention module is placed in the decoder part and is divided into channel attention and spatial attention to allow the model to capture the respective relevance from the channel attention and spatial attention. In a lung and esophageal dataset, the model achieved higher IOU than the U-Net and U-Net++ did. However, the computing speed, GPU memory, and gradient descent may be affected when there are more parameters.

#### 2.1.6. UNet++

The main difference between U-Net++ (Figure 4e) and U-Net in terms of architecture is the design of the skip connection [22,23]. In U-Net++, instead of a long skip connection, the space between the encoder and the decoder is filled with dense convolutional blocks, as shown in Figure 2. The features transmitted between those convolutional blocks are like a decoder–encoder and in a single direction. Meanwhile, the skip connections are connected horizontally between these convolutional blocks, crossing up to three convolutional blocks at most. All these skip connections are horizontal and transmitted from the encoder to the decoder.

The computation of a convolutional block includes convolutions and activation functions. If a convolutional block is not on the far-left side or at the same location as the encoder in a traditional FCN (*j* > 0), the block will receive additional upsampling information from the convolutional block in the lower layer. With this arrangement, the neural network is akin to a combination of one encoder and multiple decoders. The equation is as follows (Z. Zhou et al. 2018 [23]):(8)$$x^{i,j}=\left\{\begin{array}{c} H\left(x^{i-1.j}\right),\;i=0\\ H(\left[\left[x^{i,k}{]}_{k=0}^{j-1},\;U\left(x^{i+1.j-1}\right)\right]\right),\;j>0 \end{array}\right.$$

Another difference between the U-Net++ and U-Net is deep supervision, which enables the model to operate in two different modes: accuracy mode and fast mode [23]. In both modes, the network generates an output (*x*^(0,*i*)) for each convolutional block in the top layer. In the accuracy mode, all outputs are averaged as the final output. In the fast mode, the model only selects one of the outputs as the final output. The selection is based on the speed of calculation and the size of the network selected by the user. This indicates that the network of the U-Net++ can be different between the training phase and the inference phase. The inference model can be pruned from the trained model according to a user’s needs. Deep supervision combines two loss functions—binary cross-entropy and the Dice coefficient—and is expressed as the following equation:(9)$$L\left(Y,\;\hat{Y}\right)=-\frac{1}{N}\overset{N}{\underset{b=1}{\sum }}\left(\frac{1}{2}\cdot Y_{b}\cdot log\hat{Y_{b}}+\frac{2\cdot Y_{b}\cdot \hat{Y_{b}}}{Y_{b}+\hat{Y_{b}}}\right)$$
where *Y* and $\hat{Y}$ represent the actual and predicted probability distribution after flattening, respectively, and N represents the batch size. To sum up, U-Net++ has an encoder–decoder network filled with dense convolutional blocks, while different skip connections cross between the encoder–decoder network and these convolutional blocks so that the gradient flow could be improved; finally, deep supervision is achieved.

U-Net++ has been widely applied in several medical domains. Zhou et al. [22,23] performed a series of experiments using the U-Net++, including histopathological, clinical–medical, and endoscopic imaging. The IOU of the U-Net++ exceeds that of the U-Net by 3% to 5%, which is not affected by using various backbones (e.g., VGG, ResNet, and DenseNet). U-Net++ contains more parameters than U-Net, but it can decrease memory usage to speed up computation by sacrificing a little accuracy through deep supervision, which still makes it a valuable network.

### 2.2. Other FCNs

#### DeepLab

DeepLab (Figure 5a) was first proposed by [24]. The network consists of two parts: atrous convolution and a conditional random field (CRF). They offer solutions to two problems. The first problem is the gradual loss of features through the downsampling process, which can be dealt with using atrous convolution, as mentioned above.

The other problem is maintaining the spatial stability to detect the centers of objects, which conflicts with the spatial accuracy of the deep convolutional neural network. The CRF combines the local features with classifiers to reinforce the edge detection ability of the model.

It has been claimed that DeepLab possesses three benefits: speed, accuracy, and simple architecture. DeepLab has evolved over time: v1, v2, v3, and, currently, v3+. Atrous spatial pyramid pooling (ASPP) (Figure 5b) is included in v2, in which features are stacked together after being filtered through atrous convolutional layers of different sizes. At present, this method not only allows different levels of features to be retained but the center of an object can be determined effectively as well [25].

In v3, the authors reviewed several common methods of semantic segmentation, such as encoder–decoder, pyramid network, and ASPP in v2. The entire architecture is readjusted first by deepening the convolutional layers, which are similar to ResNet, so as to obtain features near the image level. To ensure that contextual information will not be lost through striding in repeated convolution layers, atrous convolution is used instead when output_stride reaches a certain level. The authors then reviewed ASPP by applying the aforementioned method of increasing the layer number horizontally in the ASPP. A feature map is then concurrently extracted through convolutions with different sizes and rates, concatenated, and then passed through a 1 × 1 convolution before output. This method increases the mIOU to 70–80% [26]. The output_stride is associated with accuracy but takes up a considerable amount of computing resources.

DeepLab also possesses remarkable edge detection capabilities in medical images. Tang et al. performed a two-stage method to accomplish liver segmentation in abdominal CT scans [27]. A faster RCNN was used to detect the location of the liver, and the output was then put into DeepLab for the segmentation task. This approach yielded the lowest volumetric overlap error in the two datasets.

Wang et al. applied DeepLab v3+ to detect gastric cancer in pathology slides [28]. The results were compared with those from a SegNet and a faster RCNN. The sensitivity, specificity, and Dice coefficient (DC) were all over 90%; in addition, the DC from DeepLab v3+ was higher than that of those methods by 12%. Ahmed et al. [29] compared the performance in detecting breast cancers from mammography between DeepLab and mask-RCNN. Following the preprocessing step, in which muscles were filtered, both models achieved an area under the curve (AUC) of more than 0.95 and a mean average precision of 0.8 and 0.75, respectively. Therefore, the authors suggest that both models can be used to assist radiologists in effectively identifying breast cancer lesions through mammography.

## 3. Data Augmentation

### 3.1. Traditional Data Processing

Data augmentation (Figure 6) is one of the common methods of reinforcing model performance in deep learning. It allows machines to generate images like the original data, but independent of it—in the model’s view—so as to be used in training. Data augmentation increases the variety of data and reduces the gap between training and validation data to improve model robustness [30]. Rigid and non-rigid transformations are often used in data augmentation. The former includes shifting, rotation, and flipping; the latter includes scaling and shearing.

In medical imaging, rigid transformation is more often used for segmentation tasks than classification tasks. Because of class imbalance, it is better to retain the original image features for model convergence. The pattern of features might be changed in non-rigid transformations, which may lead to poor training results, especially with a small amount of data. In addition, if the left and right sides of an image should be detected by the model, flipping may confuse the model.

Owing to rare data, data augmentation is quite important in medical imaging training tasks. However, there is a paucity of studies that focus on the contribution of data augmentation [31,32,33]. 

### 3.2. Generating Data from Models

Another way to augment data is to use deep learning models. A model is trained to generate new data, which are not only transformed from the original data but also the corresponding annotations. Two methods are discussed here: generative adversarial network (GAN) and gradient-weighted class activation mapping (Grad-CAM).

#### 3.2.1. Generating Data from GAN

A GAN consists of a generator network and a discriminator network. The aim of the generator network is to generate an image that is hard for the discriminator to differentiate from the real image, while the aim of the discriminator network is to differentiate the generated image from the real image. Through the adversarial process, the generator is able to create images that look almost real, and the discriminator is also able to detect the subtle difference between the real and generated images [34].

GANs have been applied in medical imaging tasks, including classification [35] and segmentation [36,37,38]. GAN can also be used in data augmentation. Maayan et al. [35] used GAN to generate CT images to improve data volume. The accuracy rose to 0.85, which was higher compared with typical data augmentation methods (0.75–0.8). Veit et al. [38] applied GAN to data augmentation to generate non-contrast CT images based on real-contrast CT images. The performance of the U-Net in segmentation rose from 0.916 to 0.932 (as represented by the Dice score).

There are some limitations to GAN, the biggest of which is convergence. The balance between the generator and the discriminator is difficult to maintain. For example, if the discriminator is overfitted, the generator will have difficulty obtaining its convergence [39]. Therefore, this may affect the quality of the data generated by the model. If the generated images have totally different features or patterns compared with the original data, the purpose of the data augmentation will be lost.

#### 3.2.2. Generating Data from Grad-CAM

It has been a question since deep learning techniques first developed as to how neural networks actually learn. People have tried to figure out the answer by developing methods such as visualizing convolutional networks [40] or based on a gradient point of view [41]. The integrated architecture is an approach called Grad-CAM. A corresponding classification image, also known as a type of heatmap, is derived from the product of computing the weight and the above activation, passing through the ReLu activation of the last convolution layer [42]. This heatmap effectively displays the object to be classified. Therefore, neural networks are perceived to be able to discriminate according to “human thoughts”. From another perspective, Grad-CAM is more similar to judgment rules or the domain knowledge of humans, though the thinking process in the background might not necessarily be similar to that of a human.

The property of the heatmap in Grad-CAM, which matches targets, can be used to help annotate in segmentation. Grad-CAM is used to obtain the approximate location and size of an object, followed by manual modification, which makes annotation more efficient [43].

## 4. Clinical Datasets and Relevant Studies

### 4.1. Lung Lesions

Some chest lesions have been studied, including lung lesions such as pneumothorax, lung nodules, pneumonia, cardiac lesions such as ventricle hypertrophy, and bony lesions such as rib fractures. Except for rib fractures, there are obvious targets for segmentation in these lesions, along with open datasets [44,45,46] for constructing a pretrained model so that the training for those tasks is more likely to succeed.

Table 2 shows research on lung lesion segmentation. Singadkar et al. [47] applied an FCN-based neural network, combined with residual blocks in the decoder section and long skip connections between the encoder and decoder, for lung nodule segmentation in CT scan images. They successfully reached an average Dice score of 0.95 and a Jaccard index of 0.887. Abedalla et al. [8] utilized multiple U-Net models with different backbones in each network for training and used a method similar to ensemble learning, in which four models are first summated according to fixed weights and then subjected to a threshold in order to accomplish segmentation via a pneumothorax during in inference phase. The weights and thresholds are manually adjusted. The network achieved a DSC of 0.86 in the 2019 Pneumothorax Challenge dataset.

### 4.2. Brain Lesions

Brain lesion detection includes brain tumors, strokes, traumatic brain injuries, and brain metastases. BraTS [56] (Figure 7a) is a brain tumor dataset with labels not only the location and size but also the cell type of tumors, primarily low-grade and high-grade gliomas. Magnetic resonance imaging (MRI) scans are divided into pretreatment and posttreatment images. In addition, each patient is scanned via instruments with varying magnetic field intensities (1.5 and 3 T) and protocols (2D and 3D). There are four major types of MR images: T1, T1c, T2, and FLAIR. The tumor edge is difficult to identify in segmentation tasks because of infiltrations, particularly those of high-grade gliomas, and the variety of degrees of contrast enhancement across different MRI scans.

Table 3 shows research on segmenting brain lesions. Isensee et al. [16] attempted to modify the structure of the U-Net architecture by using batch normalization and short skip connections such as s residual block in ResNet instead of a traditional convolutional block. Finally, they summated the outputs of each layer in the ascending part before entering the output part. The Dice coefficient was superior to that of the traditional U-Net architecture. In summary, most of the leading models in the BraTS dataset over the years have been based on U-Net architecture. Some of them have been modified from convolutional blocks, while others have been adjusted at the ascending part.

The intensity of stroke lesions in CT images can change over time after examination, especially infarction strokes. [59] In addition to CT, MRI datasets such as ISLES [60] have been established in recent years. Models trained with those datasets not only determine the location and region of stroke lesions but also facilitate physicians to determine the severity of brain damage and may predict the prognosis and potential of recovery. Zhang et al. [61] developed a multi-plane neural network structure to segment stroke lesions from diffusion-weighted magnetic resonance images. In contrast with the direct usage of 3D neural networks, they applied three neural networks that correspond to images on three different planes—axial, coronal, and sagittal—then integrated them into a multi-angle neural network, which is called a multi-plane fusion network. This neural network offers both segmentation and detection functions and can retain the original information from the input. Based on images from three different planes, the edges of lesions can be identified more accurately. The authors achieved a Dice coefficient of 62.2% and a sensitivity of 71.7% in the ISLES dataset.

**Table 3 diagnostics-12-02765-t003:** Research on brain lesion segmentation.

Authors (Year)	Method	Medical Image	Performance	Notes
*Brain tumor*				
Havaei et al. [62] (2016)	Deep CNN	Magnetic resonance images	DC ^1^: 0.88	Cascade architecture using pre-output concatenation
Pereira et al. [63] (2016)	CNN-based	Magnetic resonance images	DC: 0.88	Patch extraction from an image before entering the CNN
Isensee et al. [16] (2018)	3D U-Net	Magnetic resonance images	DC: 0.85	Modified from U-Net; summation for multi-level features
Xu et al. [33] (2020)	U-Net	Magnetic resonance images	DC: 0.87	Attention-U-Net
McKinley et al. [64] (2018)	deepSCAN	Magnetic resonance images	Mean DC ET ^2^: 0.7 WT ^3^: 0.86 TC ^4^: 0.71	Bottleneck CNN design; dilated convolution
*Stroke*				
Wang et al.[65] (2016)	Deep Lesion Symmetry ConvNet	Magnetic resonance images	Mean DSC ^5^: 0.63	Combined unilateral (local) and bilateral (global) voxel descriptor
Monteiro et al. [66] (2020)	DeepMedic	Computed tomography	Differs according to size	Three parallel 3D CNNs for different resolutions
Zhang et al. [61] (2020)	U-Net	Magnetic resonance images	DSC: 0.62 IoU ^6^: 0.45	FPN for extraction first

^1^ Dice coefficient, ^2^ enhanced tumor, ^3^ whole tumor, ^4^ tumor core, ^5^ Dice similarity coefficient, ^6^ intersection over union.

### 4.3. Abdomen

#### Abdominal Organ Segmentation

The solid organs in the abdomen such as the liver, kidneys, spleen, and pancreas, as well as lower abdomen organs such as the prostate, have more prominent edges and distinct intensity values compared with the background, which is usually fat or peritoneum. Thus, they are obvious targets for segmentation. Convincing results could be achieved with traditional computer vision techniques [67,68]. Regarding the urinary bladder, due to its prominent edges, despite being a hollow organ, segmentation tasks could still be accomplished with trained models (particularly in the case of a distended bladder) [69]. There is a wealth of data focusing on abdominal organ segmentation [70,71,72]. In recent years, the application of deep learning for segmentation tasks has been considerably robust [73,74].

## 5. Abdominal Lesion Segmentation

Relevant studies on abdominal tumor segmentation mostly focus on the liver, kidneys, prostate, and urinary bladder. Manjunath et al. [75] trained a U-Net model with ResNet as the backbone to segment the liver and liver tumors from CT images and achieved a DSC of 96.35 and an accuracy of 99.71% in the liver tumor segmentation (LiTS) dataset [58] (Figure 7b). Vorontsov et al. [76] developed an FCN network trained from LiTS and a local dataset and made inferences to segment metastatic liver lesions from colorectal cancers in another set of CT scan images from several local hospitals. Despite experiencing limitations such as small data volume and the wide spread of patient origins, the model achieved a positive predictive value of over 80% for lesions larger than 1 cm. Liang et al. [77] applied a square-window CNN to segment pancreatic cancers. The model was trained from the multi-phased MRIs of 27 patients, with extraction and augmentation from 245,000 normal and 230,000 abnormal patch images. The model was tested on the images of 13 patients, and it achieved a DSC of 0.73, thus solidifying it as an assistive means for radiation oncologists to ascertain the gross tumor volume. Pellicer-Valero et al. [78] purposed a Retina U-Net trained from two MRI datasets to analyze the prostate and tumors and to predict their Gleason grade group. Chen et al. [79] applied the 3D AlexNet for prostate cancer segmentation and achieved an accuracy of 0.92 and a DSC of 0.977.

In 2016, Cha et al. [80] conducted a pilot study to evaluate whether deep-learning neural networks can segment urinary bladder tumors in CT images and further assess the treatment outcomes. The results showed that the assessment of tumor size changes from the model was not inferior to those derived from humans according to the World Health Organization (WHO) and the Response Evaluation Criteria in Solid Tumors (RECIST) guidelines [81]. The authors subsequently conducted a study that combines radiomic feature analysis [82], whereby lesions are segmented by the deep learning model to determine changes in tumor size in contrast-enhanced CTs before and after treatment. Conclusively, the method offers higher accuracy for determining tumor volume, but it is not necessarily superior to human judgment for determining treatment efficacy. Nonetheless, the model can be used as an effective tool for radiologists in diagnosis. Table 4 summarizes the research on the segmentation of abdomens mentioned above.

Physicians choose different tools for examination according to the location of tumors and, therefore, obtain quite different images from traditional image scopes. For example, endoscopy is used for gastrointestinal tumors. Unlike CT or MRI images presenting the lesion from a general anatomy point of view, endoscopic images only present the lesion in the regional field via the scope passing through the lumen, such as when using endoscopic ultrasound (EUS) to diagnose pancreatic tumors. The endoscopic images are generated in real-time, so they are like streams instead of slices of images. Physicians can still get snapshots that they are interested in, but, in general, it has no regular interval between the snapshot images as CT or MRI images do. For this reason, it is better to perform tumor segmentation in real-time, as with the stream, where the results are displayed to the physician. Li et al. [83] trained a U-Net to segment EUS-observed GIST tumors, which are derived from the gastric stroma. The authors made some modifications to the model to address the problems of shadowing and size differences, which often appear in EUS imaging. The model achieved a DSC of over 0.9 in the test data. Tonozuka et al. [84] used a U-Net to segment pancreatic tumors in EUS and achieved a median IoU of 0.77. The results were affected by unclear tumor margins but not by respiratory movements.

**Table 4 diagnostics-12-02765-t004:** Research on abdomen segmentation.

Authors (Year)	Method	Medical Image	Performance	Notes
*Abdominal organ*			Differs between organs	
Landman et al. [85] (2018)	FCN	Magnetic resonance images	DSC ^1^: 0.56–0.93	
Gibson et al. [86] (2018)	Dense V-Networks	Computed tomography	DC ^2^: 0.76–0.9	High-resolution activation maps; batch-wise spatial dropout
Kim et al. [73] (2020)	3D U-Net Atlas-based	Computed tomography	DC: 0.60–0.96 DC: 0.15–0.81	Multi-organs were tested; the U-Net result could be comparable to that of an interobserver
Kart et al. [74] (2021)	nn-UNet	Magnetic resonance images	DC: 0.82–0.9	
*Abdominal tumor*				
Abdel-Massieh et al. [87] (2010)	CV ^3^ method	Computed tomography	Overlap error: 0.22	Gaussian blurring; isodata threshold
Abd-Elaziz et al. [88] (2014)	CV method	Computed tomography	Error rate: 0.002–0.012	Regional pixel growing and morphological processing
Manjunath et al. [75] (2021)	ResUNet	Computed tomography	DSC: 0.96	Replacing convolutional blocks with residual blocks
Vorontsov et al. [76] (2019)	FCN	Computed tomography	DSC per lesion: (automated) <10 mm: 0.14 10–20mm: 0.53 >20 mm: 0.68	Two-step segmentation: the first is FCN for livers, and the second is FCN for lesions in livers
Liang et al. [77] (2020)	Square-window based CNN	Magnetic resonance images	DSC: 0.73 on the test set	
Pellicer-Valero et al. [78] (2021)	Retina U-Net	Magnetic resonance images	DSC: (prostate) 0.915	Two 3D CNNs: the first one takes a T2-weighted MRI as the input, and the second one takes an MRI and the output from the first one as inputs
Chen et al. [79] (2020)	3D AlexNet	Magnetic resonance images	DSC: 0.97	
Li et al. [83] (2020)	MRBS-U-Net	Endoscopic ultrasound	DSC: 0.92	

^1^ Dice similarity coefficient, ^2^ Dice coefficient, ^3^ computer vision.

## 6. Discussion

There have been major advancements in the application of deep learning for medical imaging segmentation since the development of the U-Net architecture. Meanwhile, FCN-related studies and techniques are making the model more robust. We think there are two major points to make sure the model is successfully trained:

(1) Data. Sufficient data are necessary for model convergence. In addition, well-defined task objectives lead to clear data labeling, while domain knowledge is critical to maintaining label consistency. To prevent class imbalance, labeled data should be sufficient and should not be skewed toward a particular class [89]. However, class imbalance is a common problem in semantic segmentation. U-Net may overcome this problem [6]. While researchers have investigated other models or approaches to deal with data problems, the best way is still to ensure the data structure [90,91,92], which requires the assistance of data scientists or software to evaluate the distribution of the class of the label. The data quality should also be involved. To achieve this, the production of data should be consistent, and unnecessary interferences and irrelevant information should be removed so that the model will not learn from the information that has nothing to do with what the model should learn during training.

(2) Model type. Due to the nature of segmentation tasks, full convolution networks are currently the solution. Edge detection of the object may be a key point for future developments. In addition to model architecture, loss function is also important. The literature has shown that, in addition to model structure modifications, adjustments of loss function have also been used to raise the performance of the model [22,23].

On the other hand, from the perspective of clinical medicine, there are still some obstacles to research:

(1) Poor data volume. Patients are a relatively small population compared with the whole human population. In addition, data security and privacy policies lead to insufficient data influence, making it challenging to establish an effective dataset.

(2) Inconsistency in data quality between medical institutes. Due to differences between examination procedures, instruments, and image processing methods, the images generated by those institutes may vary in terms of contrast, brightness, or resolution. These issues may occur between and even within medical institutes.

(3) The low reliability of machine-assisted diagnostic tools is perceived by clinical workers. The results of previous research suggest that making image-based diagnoses using deep learning models is not inferior to experienced physicians or experts in certain circumstances. Moreover, the model may even effectively compensate for less-experienced clinicians [93]. Nevertheless, there are different procedures and protocols in each medical department, which results in two issues: Whether a model can be seamlessly integrated into an environment, and whether the system can be acceptable with an inserted model. The solutions to these issues hinge on the collaboration between medical professionals, information professionals, and researchers. The inclusion of multidisciplinary professionals is crucial for expediting the process.

One of the directions for future developments in semantic segmentation is federated learning, which was initially an approach proposed by Google for smartphone keyboard input prediction [94]. Numerous medical departments can train the model on their own dataset to contribute weights to the final model. The data do not need to pass through a platform, so the patient information is secure, and the model is able to obtain diverse and large data volumes as well. However, there may be some problems. One of them is the variety of data volumes. Medical institutes that can provide larger data volumes would make greater contributions to the model. If the gap in data volumes between institutes is too large, such that most of the data are provided from one or only a few institutes, the model will have a similar or even worse performance than those only trained on their own dataset, which will lower their willingness to participate. Another problem is cost-effectiveness. The weights are contributed by each medical institute, and how the gradient is backpropagated significantly affects the performance. By using general averaging methods, hospitals possessing higher data volumes may contribute more weights, which results in a model that is biased toward hospitals, affecting the model’s generalization. By using other methods, it may benefit only those providing fewer data instead. Since those institutes owning larger data volumes will be more likely to make larger investments in maintaining the dataset, the cost-effectiveness problem will become a key concern. The differences between the imaging qualities of each medical institute will also raise concerns about the performance of generalization [95]. Thus, there are still some obstacles needed to be overcome in federated learning.

## 7. Conclusions

In this review, we reviewed several FCN models for their applications in medical image segmentation. There have been an increasing number of advancements since the development of U-Net: some frameworks deal with blurred objects, and other frameworks are good at detecting object burdens.

Despite the lower accessibility of medical data, they are still applicable to train models with critical data augmentation. We summarized two main skills for data augmentation: figure transformation and the GAN network. Although Grad-CAM may not generate figures from original images, it can assist in labeling by generating heatmaps of images.

Furthermore, we also reviewed studies related to model performance in clinical datasets. Generally, the proposed models showed good performance in testing datasets, and even in some clinical images collected by researchers. However, model performance remains a challenge after deployment. To overcome the gap between the dataset performance and clinical practice, more local data should be collected to train new models or to perform transfer learning with pre-trained models. In addition, it is necessary to constantly improve the model through cooperation with data scientists, data analysts, and clinical practitioners.

## Figures and Tables

**Figure 1 diagnostics-12-02765-f001:**
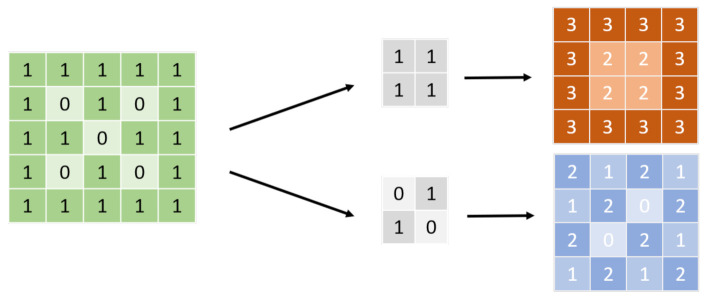
CNN kernel. Before input, the figure will be transformed into a signal, as shown on the left; 0/1, for example. The arrays in the middle are called the convolutional kernel. The size of the kernel must not be larger than the input. The neural network applies several kernels with different weight compositions to the input to obtain feature maps, which are usually dot products, as shown on the right. The neural network extracts a feature that can make them accomplish the tasks from those kernels.

**Figure 2 diagnostics-12-02765-f002:**
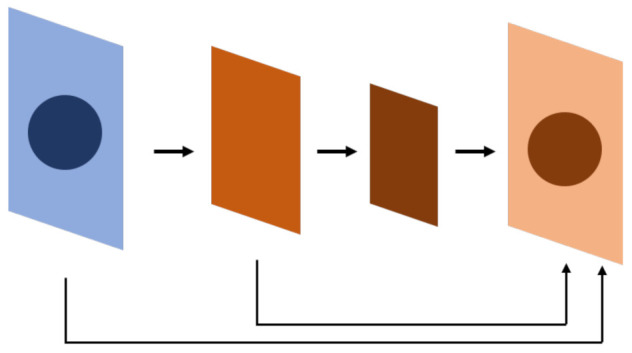
FCN. The whole neural network is composed of a convolutional neural network, which is different from a conventional neural network, especially on the bottom 1~2 layers. The result is at a pixel level, so the network can afford a segmentation task. FCN uses a convolutional neural network to extract features and uses skip connections to find out the location of the feature at the whole-map level.

**Figure 3 diagnostics-12-02765-f003:**
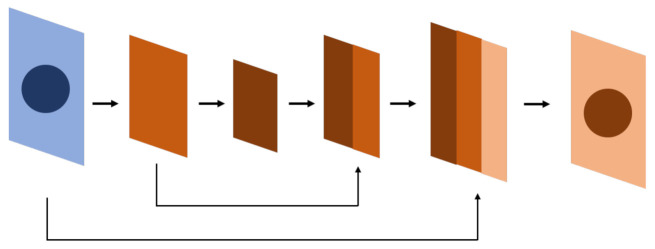
U-Net, derived from FCN. The result from the FCN experiment shows that the inference will be more precise if the same level feature before downsampling is summed before the feature goes into upsampling. U-Net applies concatenation instead of summation; however, the summation is also used in some research.

**Figure 4 diagnostics-12-02765-f004:**
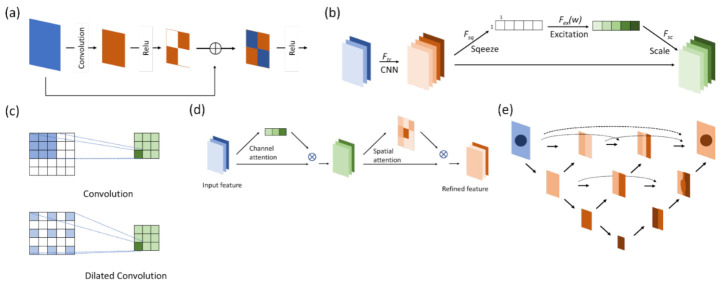
The blocks or modules used in different neural networks for computer vision deep learning. (**a**) The residual block applies a skip connection to keep the gradients from vanishing or being lost during backpropagation. (**b**) The SE module assigns weights to each channel for representation and multiples them to the corresponding feature map to increase the robustness. (**c**) Dilated convolution expands the reception field to enhance its ability to process objects with blurred borders. (**d**) The attention module identifies the relationships between regional features so that it detects object borders more effectively. (**e**) U-Net++ block-fills the space between the encoder and the decoder with convolutional blocks to improve performance.

**Figure 5 diagnostics-12-02765-f005:**
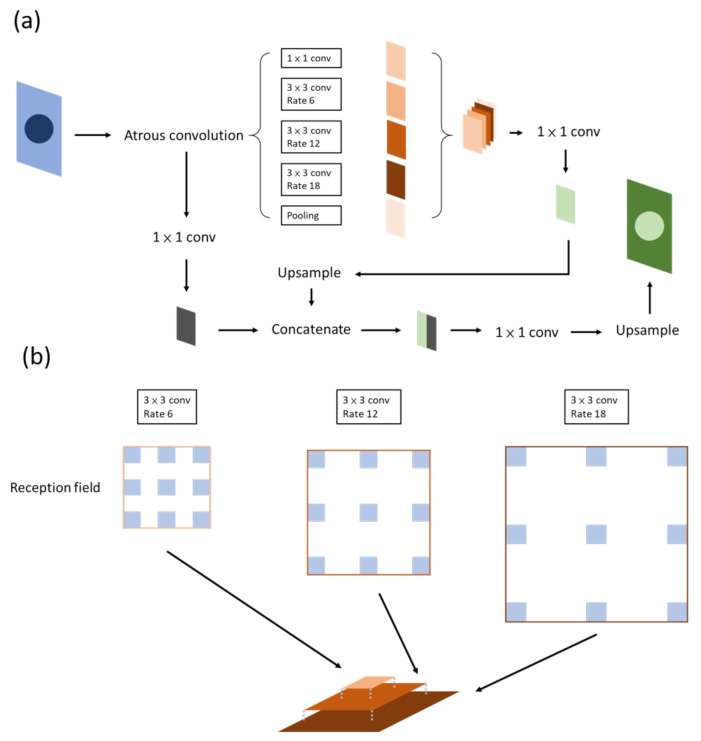
Deeplab. (**a**) Version v1; the critical part of the network is the ASPP, or atrous convolution, which is shown in (**b**).

**Figure 6 diagnostics-12-02765-f006:**
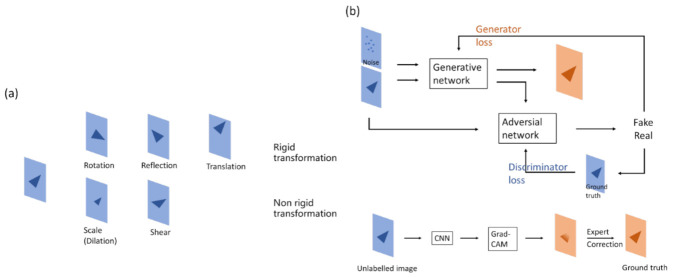
Method for data augmentation. (**a**) Making changes to figures, the method can usually be separated as a rigid or non-rigid transformation. Rigid transformation geometrically changes the original image, while non-rigid transformation changes the size or shape with respect to the original image. (**b**) (Upper) The generative part of the GAN neural network yields figures that have similar properties, which can be used to increase the data volume. (Lower) Grad-CAM shows the heatmap of figures, which can be used for annotations, thus reducing manual workloads.

**Figure 7 diagnostics-12-02765-f007:**
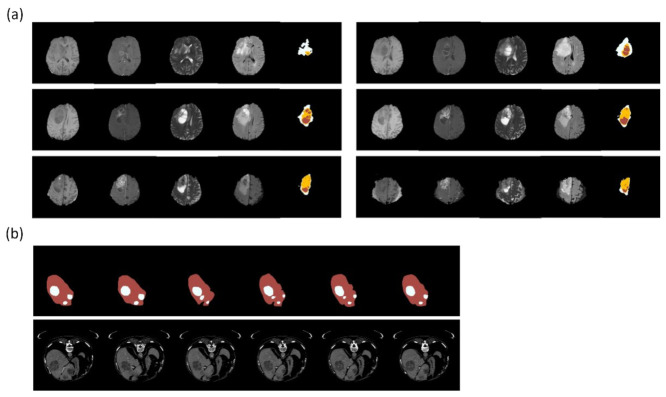
Clinical dataset for training segmentation model. (**a**) BraTS is a dataset for glioblastoma multiforme. The figures show part of a case series with different MRI weights (from the left: T1, T1ce, T2, FLAIR) and annotations (rightmost: white—perifocal edema; yellow—tumor; red—necrosis). (**b**) LiTSis a dataset about liver and liver tumor segmentation. The figures show part of a case series with annotations (upper: red—liver; white—tumor) and CT images (lower). Reference: (**a**) from BRATS (Menze et al. [25,56,57]); (**b**) from LiTS (Bilic et al. [58]). The figures were illustrated by using Python 3.6 from the datasets.

**Table 1 diagnostics-12-02765-t001:** Models for segmentation tasks.

Authors (year)	Method	Dataset	Medical Image	Performance
Ciresan et al. [7] (2012)	Sliding window convolutional network	EM segmentation challenge	Electric microscopy	Warping error: 0.000420
Ronneberger et al. [6] (2015)	U-Net	EM segmentation challenge	Electric microscopy	Warping error: 0.000353
		PhC-U373	Light microscopy	IoU: 0.92
		DIC-HeLa	Light microscopy	IoU: 0.78
Abedalla et al. [8] (2021)	ResUNet	SIIM-ACR 2019	Chest X-ray	Mean DSC: 0.86
Wang et al. [9] (2020)	SE-FCN	From their own institute	Chest X-ray	MPA:0.93
				DSC:0.92
Zhou et al. [10] (2020)	MSDU-Net	LiTS	Computed tomography	Scan angle range [0°, 150°] PSNR ^1^: 33.18 SSIM ^2^: 0.9453 UIQI ^3^: 0.9937
Yousefi et al. [11] (2021)	DDAUNet	Leiden University Medical Center, the Netherlands	Computed tomography	Mean DSC:0.79
Cai et al. [12] (2020)	MA-UNet	LUNA	Computed tomography	Lung: mIoU:0.96 Esophagus cancer: mIoU: 0.65
	UNet++			
	DeepLab (v3)			

^1^ Peak Signal Noise Ratio, ^2^ Structural Similarity Index, ^3^ Universal Image Quality Index.

**Table 2 diagnostics-12-02765-t002:** Research on lung lesion segmentation.

Authors (Year)	Method	Medical Image	Performance	Notes
Wang et al. [48] (2017)	CF-CNN	Computed tomography	DC: 0.82	Central-focused CNN: extract features from 3D and 2D simultaneously
Wang et al. [49] (2017)	MV-CNN	Computed tomography	DC: 0.77	Multi-scaled CNN
Maqsood et al. [50] (2021)	DA-Net	Computed tomography	DC: 0.81 IoU: 0.76	U-Net-based, with atrous convolution and dense connection
Meraj et al. [51] (2020)	CNN	Computed tomography	Accuracy: 0.83	For nodule detection using PCA and other machine learning techniques
Singadkar et al. [47] (2020)	DDRN	Computed tomography	DSC: 0.95	ResNet-based, with deep deconvolution (residual block at the decoder)
Zhao et al. [36] (2020)	3D U-Net	Computed tomography		3D U-Net combined with GAN for segmentation; another CNN for classifying nodule
Usman et al. [52] (2020)	3D U-Net	Computed tomography	DSC: 0.88 (consensus)	3D voxel feature, ResUNet, with semi-automated ROI selection
Keetha et al. [53] (2020)	U-Det	Computed tomography	DSC: 0.83	U-Net cooperates with a bidirectional feature network (Bi-FPN)
Ozdemir et al. [54] (2020)	3D Vnet	Computed tomography	Sensitivity: 0.97	Combined segmentation and classification for lung nodule diagnosis
Hesamian et al. [55] (2019)	FCN	Computed tomography	DSC: 0.81	Atrous convolution and residual block in FCN combined with conditioned random field (CRF)

## Data Availability

Figure 7a: BRATS. This is an open dataset; the website is as follows:http://braintumorsegmentation.org/ (accessed on 8 October 2022). Figure 7b: LiTS. This is an open dataset; the website is as follows: https://www.kaggle.com/datasets/andrewmvd/liver-tumor-segmentation (accessed on 3 October 2022).

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
