# Peer review of "Fully Convolutional Network for the Semantic Segmentation of Medical Images: A Survey"

_diagnostics, 2022, doi:10.3390/diagnostics12112765_

Round 1

Reviewer 1 Report

I have read the manuscript “Semantic segmentation on medical image by using fully convolutional-based neural network: a survey” by Sheng-Yao Huang et al. The topic is worth interesting and deserves a dedicated review to highlight the advantages and disadvantages of using FCN with respect to other techniques (including different kinds of deep learning approaches) for medical imaging segmentation by using artificial intelligence and related approaches. However, the paper is not well written and cannot be published in the present version: it is full of English and stylist errors (already in the title),  does not provide a complete introduction to the topic (e.g. what is a neural network?) and does not contain any medical images to show the possible different outcomes of the considered approaches.

Therefore, in my opinion, the manuscript needs a profound revision before being considered for publication.

Author Response

I have read the manuscript “Semantic segmentation on medical image by using fully convolutional-based neural network: a survey” by Sheng-Yao Huang et al. The topic is worth interesting and deserves a dedicated review to highlight the advantages and disadvantages of using FCN with respect to other techniques (including different kinds of deep learning approaches) for medical imaging segmentation by using artificial intelligence and related approaches. However, the paper is not well written and cannot be published in the present version: it is full of English and stylist errors (already in the title), does not provide a complete introduction to the topic (e.g. what is a neural network?) and does not contain any medical images to show the possible different outcomes of the considered approaches.

Therefore, in my opinion, the manuscript needs a profound revision before being considered for publication.

We appreciate the supportive comments. We have revised the manuscript accordingly on the basis of your comments and suggestions.

Point 1: The paper is not well written and cannot be published in the present version: it is full of English and stylist errors (already in the title).

Response 1: Thank you for your comment. We sent the manuscript to language editing. The following is the proof. (the picture is in attached file)

We revised the title: Now, it is

Fully convolutional network for semantic segmentation on medical image: a survey

We also revised the abstract. Please see the following text.

Revision (p. 1, line 13 -33):

There have been major developments in deep learning in computer vision since 2010s. Deep learning has contributed to the wealth of data in medical image processing, and semantic segmentation is a salient technique in this field. This study retrospectively reviews recent studies on the application of deep learning for segmentation tasks in medical imaging and proposes potential directions for future development, including model development, data augmentation processing, and dataset creation.  The strengths and deficiencies of studies on models, data augmentation, and their application to medical image segmentation were analyzed. Fully convolutional network developments have led to the creation of the U-Net and its derivatives. Another noteworthy image segmentation model is DeepLab. Regarding data augmentation, due to the low data volume of medical images, most studies focus on the means to increase the wealth of medical image data. Generative adversarial networks (GAN) increase data volume via deep learning. Despite the increasing types of medical image datasets, there is still a deficiency of datasets on specific problems, which should be improved moving forward. Considering the wealth of ongoing research on the application of deep learning processing on medical image segmentation, the data volume and practical clinical application problems must be addressed to ensure that the results are properly applied.

There are changes made at manuscript to correct grammar and spelling. Please see the revised manuscript.

Point 2: The paper does not provide a complete introduction to the topic (e.g. what is a neural network?)

Response 2: Thank you for your comment. We added introduction for concolutional neural network (CNN) and fully convolutional network (FCN), as follows:

Revision:

(for CNN, p.2, line 71-74)  Fig. 1 CNN kernel. Before input, the figure will be transformed into signal, as showing on the left, 0/1 for example. The arrays in the middle are called convolutional kernel. The size of kernel must not be larger than input. The neural network applies several kernels from different weight composi-tion to the input to get feature maps, which are usually dot products, as showing in the right. The neural network extracts feature that can make them accomplish the tasks from those kernels.

(for FCN, p.2, line 75-79)  1. Fully convolutional network (FCN)

One of the most selected models for segmentation tasks consist of FCN (Fig. 2). The difference between FCNs and traditional convolutional neural networks is that the last layer is not a fully connected layer that enables the model to integrate information. Instead, the final number of output channels is modified through convolutional networks. The main benefit of this approach is that the model has no restrictions from the full connection layer so that the size of input can be flexible.

However, there is a gap between CNN an FCN. We didn’t explain the weak point of CNN in segmentation task, which is the reason to introduce FCN. We added a paragraph to fill up the gap.

Revision: (Introduction, p.2, line 57-67)

 …However, in traditional CNN models, there are limitations of segmentation. One of them is about the features extracted from the models. When using a smaller kernel, the features will become more local to original image. The global information, such as location, may be lost. However, when using a larger kernel, the context of feature may decrease6. Another limitation is about the data availability of biomedical segmentation tasks. Due to the privacy and medical profession, medical data volume is often small, when compared to the data volume in other fields6. New models have been developed to solve these problems. Another solution is data augmentation.

In this article, we will discuss the models proposed for semantic segmentation, data augmentation techniques, and their current application on clinical data.

Point 3: The paper does not contain any medical images to show the possible different outcomes of the considered approaches.

Response 3: Thank you for the comment. There are different outcomes present in references. The ideal way is to parallely present the result showing in those research in the manuscript. We’ve contacted the journal publisher for request their permission of using the figures, yet I didn’t get the response. Also, the figures in different research might have been edited before published, so it would be difficult to compare them in figures. We make comparison via the tables in the manuscript with the result (e.g. dice coefficient) as reference.

We found some of the dataset mentioned in the manuscript, showing part of the images and annotations in figures as representation. The revision is as follows.

Revision (p.13, line 421-428): (the figure is in attached file)

Figure 7. Clinical dataset for training segmentation model. (a) BRATs is a dataset about glioblastoma multiforme. The figures show a part of a case series with different MRI weights (from left: T1, t1ce, t2, flair) and annotations (the most right, white: perifocal edema, yellow: tumor, red: necrosis). (b) LiTSis a dataset about liver and liver tumor segmentation. The figures show part of a case series with annotations (up part, red: liver, white: tumor) and CT images (low part). Reference: (a) from BRATS (Menze et al. 50), (b) from LiTS (Bilic et al. 51)

Reviewer 2 Report

The authors present a survey on the semantic segmentation of medical images with the use of fully convolutional neural networks. They included in their study recent forks, primarily from the period 2019-2021, which cover the fully convolutional networks, modified U-Net models and other FCNs. The problem of data augmentation was also investigated, along with the present clinical datasets - lung lesions, brain lesions, abdominal organ and lesion segmentation. Analyzed results have been widely discussed.

The following recommendations would increase the quality of the draft, if taken into account:

- The text in Fig. 4 and 6 is small and should be enlarged;

- A few references in the field from earlier developments, prior 2019, could be included in the study for comparison purposes, placing the qualities of the methods in the present tables;

- There should be a distinct conclusion in a separate section at the end of the paper, which summarizes the main observations from the survey, long enough to be comprehensive.

The paper could be accepted for publication after minor revision.

Author Response

The authors present a survey on the semantic segmentation of medical images with the use of fully convolutional neural networks. They included in their study recent forks, primarily from the period 2019-2021, which cover the fully convolutional networks, modified U-Net models and other FCNs. The problem of data augmentation was also investigated, along with the present clinical datasets - lung lesions, brain lesions, abdominal organ and lesion segmentation. Analyzed results have been widely discussed.

The following recommendations would increase the quality of the draft, if taken into account:

- The text in Fig. 4 and 6 is small and should be enlarged;

- A few references in the field from earlier developments, prior 2019, could be included in the study for comparison purposes, placing the qualities of the methods in the present tables;

- There should be a distinct conclusion in a separate section at the end of the paper, which summarizes the main observations from the survey, long enough to be comprehensive.

The paper could be accepted for publication after minor revision.

We appreciate the supportive comments. We have revised the manuscript accordingly on the basis of your comments and suggestions.

Point 1: The text in Fig. 4 and 6 is small and should be enlarged.

Response 1: Thank you for your comment. We adjusted the font size of text in Fig. 4 and Fig. 6 (from 18 to 24). Also, we added the text for Fig.4 and 6 for more details (as follows) , making connection between the figures and main article closer. Please see the new figure we present in the manuscript.

Revision:

(p.8, line 281-287)

Figure 4. The blocks or modules used in different neural network on computer vision deep learning. (a) Residual block applies skip connection to keep the gradients from vanishing or loss during back propagation. (b) SE module assigns weights to each channel for representation and multiples them to the corresponding feature map to increase the robustness. (c) Dilated convolution expands the reception field to enhance its ability to process objects with blurred borders. (d) Attention module identifies the relationships between regional features so that it detects object borders more effectively. (e) Unet++ block fills the space between the encoder and the decoder with convolutional blocks to improve performance.

(p.12, line 394-401)

Figure 6. Method for data augmentation. (a) Making changes on figures, the method usually can be separated as rigid and non rigid transformation. Rigid transformation geometrically changes the original image, while non rigid transformation changes the size or shape with respect to the original image. (b) (Upper) The generative part of the GAN neural network yields figures that have similar properties, which can be used to increase the data volume. (Lower) Grad-CAM shows the heat map of figures, which can be used for annotations, thus reducing manual workloads.

Point 2: A few references in the field from earlier developments, prior 2019, could be included in the study for comparison purposes, placing the qualities of the methods in the present tables.

Response 2: Thank you for your kindly suggestions. We added references prior to 2019 in each tables, corresponding to different subtitles.

Revision:

(table 1: Ciresan et al. (2012), table 2: Wang et al. (2017), table 3: Pereira et al. (2016), Wang et al. (2016), table 4: Landman et al. (2018), Gibson et al. (2018), Abdel-massieh et al. (2010), Abd-Elaziz et al. (2014)) Please see the renewed tables. The references are also listed here:

  1. Cires¸an DC, Giusti A, Gambardella LM, Urgen Schmidhuber J¨. Deep Neural Networks Segment Neuronal Membranes in Electron Microscopy Images.; 2012. http://www.idsia.ch/
  2. Wang S, Zhou M, Liu Z, et al. Central focused convolutional neural networks: Developing a data-driven model for lung nodule segmentation. Med Image Anal. 2017;40:172-183. doi:10.1016/j.media.2017.06.014
  3. Wang S, Zhou M, Gevaert O, et al. A multi-view deep convolutional neural networks for lung nodule seg-mentation. In: 2017 39th Annual International Conference of the IEEE Engineering in Medicine and Biology Society (EMBC). IEEE; 2017:1752-1755. doi:10.1109/EMBC.2017.8037182
  4. Pereira S, Pinto A, Alves V, Silva CA. Brain Tumor Segmentation Using Convolutional Neural Networks in MRI Images. IEEE Trans Med Imaging. 2016;35(5):1240-1251. doi:10.1109/TMI.2016.2538465
  5. Wang Y, Katsaggelos AK, Wang X, Parrish TB. A deep symmetry convnet for stroke lesion segmentation. In: 2016 IEEE International Conference on Image Processing (ICIP). IEEE; 2016:111-115. doi:10.1109/ICIP.2016.7532329
  6. Landman BA, Bobo MF, Huo Y, et al. Fully convolutional neural networks improve abdominal organ seg-mentation. In: Angelini ED, Landman BA, eds. Medical Imaging 2018: Image Processing. SPIE; 2018:100. doi:10.1117/12.2293751
  7. Gibson E, Giganti F, Hu Y, et al. Automatic Multi-Organ Segmentation on Abdominal CT With Dense V-Networks. IEEE Trans Med Imaging. 2018;37(8):1822-1834. doi:10.1109/TMI.2018.2806309
  8. Abdel-massieh NH, Hadhoud MM, Amin KM. Fully automatic liver tumor segmentation from abdominal CT scans. In: The 2010 International Conference on Computer Engineering & Systems. IEEE; 2010:197-202. doi:10.1109/ICCES.2010.5674853
  9. Abd-Elaziz OF, Sayed MS, Abdullah MI. Liver tumors segmentation from abdominal CT images using region growing and morphological processing. In: 2014 International Conference on Engineering and Technology (ICET). IEEE; 2014:1-6. doi:10.1109/ICEngTechnol.2014.7016813

Point 3: There should be a distinct conclusion in a separate section at the end of the paper, which summarizes the main observations from the survey, long enough to be comprehensive.

Response 3: Thank you for the comment. We added a new section for conclusion. In this section, we make a brief summary to each section according to the observations. The section is at the end of the article:

Revision (p.18-p.19, line 592-609):

  1. Conclusion

In this review, we reviewed several FCN models for their application in medical image segmentation. There has been increasing number of advancements since the development of Unet : Some frameworks deal with blurred objects, and the other framework is good at detection of object burdens.

Despite the lower accessibility of medical data, it’s still applicable to train models with critical data augmentation. We summarized three main skills for data augmentation: figure transformation, GAN network. Although Grad-CAM may not generate figures from original images, it can assist in labelling by generating the heat maps of images.

Besides, we also reviewed studies related to model performance in clinical datasets. Generally, the proposed models showed good performance in testing datasets, or even in some clinical images collected by researchers. However, model performance remains a challenge after deployment. To overcome the gap between the dataset performance and clinical practice, more local data should be collected to train new models or to perform transfer learning on pre-trained models. Also, it is necessary to constantly improve the model through cooperation with data scientists, data analysts, and clinical practitioners.

Round 2

Reviewer 1 Report

I am satisfied with the authors' revisions that substantially improved the quality of their manuscript.